# Magnetic alignment enhances homing efficiency of hunting dogs

**Kateřina Benediktová[1]\*, Jana Adámková[1], Jan Svoboda[1], Michael Scott Painter[1,2], Luděk Bartoš[3,4], Petra Nováková[1], Lucie Vynikalová[5], Vlastimil Hart[1], John Phillips[6], Hynek Burda[1]\***

[1]Department of Game Management and Wildlife Biology, Faculty of Forestry and Wood Sciences, Czech University of Life Sciences, Praha, Czech Republic; [2]Biology Department, Barry University, Miami, United States; [3]Department of Ethology, Institute of Animal Science, Praha, Czech Republic; [4]Department of Ethology and Companion Animal Science, Faculty of Agrobiology, Food and Natural Resources, Czech University of Life Sciences, Praha, Czech Republic; [5]Department of Zoology and Fisheries, Faculty of Agrobiology, Food and Natural Resources, Czech University of Life Sciences, Praha, Czech Republic; [6]Department of Biological Sciences, Virginia Tech, Blacksburg, United States

**Abstract** Despite anecdotal reports of the astonishing homing abilities in dogs, their homing strategies are not fully understood. We equipped 27 hunting dogs with GPS collars and action cams, let them freely roam in forested areas, and analyzed components of homing in over 600 trials. When returning to the owner (homewards), dogs either followed their outbound track ('tracking') or used a novel route ('scouting'). The inbound track during scouting started mostly with a short (about 20 m) run along the north-south geomagnetic axis, irrespective of the actual direction homewards. Performing such a 'compass run' significantly increased homing efficiency. We propose that this run is instrumental for bringing the mental map into register with the magnetic compass and to establish the heading of the animal.

**\*For correspondence:**
benediktovak@fld.czu.cz (KB);
burda@fld.czu.cz (HB)

**Competing interests:** The authors declare that no competing interests exist.

## Introduction

Homing, broadly defined as the ability to return to a known goal location (e.g. breeding grounds, shelter sites) after displacement (*Schmidt-Koenig and Keeton, 1978*; *Papi, 1992*; *Wiltschko and Wiltschko, 1995*), has been shown in a taxonomically diverse range of vertebrates that rely on a multitude of cues, for example visual, olfactory, acoustic, celestial, magnetic, and idiothetic (*Schmidt-Koenig and Keeton, 1978*; *Papi, 1992*; *Wiltschko and Wiltschko, 1995*; *Cullen and Taube, 2017*; *Lohmann, 2018*; *Mouritsen, 2018*). However, designing systematic studies to characterize the navigational strategies and underlying sensory mechanisms mediating homing behaviour in non-migratory species, particularly in free-ranging mammals, have proven difficult, and our understanding of large-scale navigation and homing remains incomplete (*Poulter et al., 2018*; *Tsoar et al., 2011*; *Wolbers and Wiener, 2014*).

Anecdotal accounts of the impressive navigation abilities of dogs have been commonplace, maybe best exemplified in World War I when 'messenger dogs' were used as couriers to deliver sensitive information across battlegrounds (*Richardson, 1920*). Nearly a century ago, the first studies designed to examine navigational abilities in dogs were performed, revealing homing success even if displaced to unfamiliar sites (Schmid 1932, 1936 cited in *Nahm, 2015*). Decades later, a more comprehensive study observed consistent homing success in a total of 26 dogs displaced without exposure to visual cues in various geographic directions. Dogs often homed using novel routes and/ or shortcuts, ruling out route reversal strategies, and making olfactory tracking and visual piloting

unlikely. Indeed, as previous authors have suggested, shedding light on the mystery of mammalian homing may require unconventional research approaches that focus on 'unusual' senses (*Nahm, 2015*).

Hunting dogs, particularly the so-called scent hounds, have been selected over generations to detect and pursue tracks of game animals and, if not followed by the hunter, to return to the place where the pursuit started, often over distances of hundreds or thousands of meters. How dogs pinpoint the owner's location using novel routes of return in highly variable densely forested habitats remains perplexing.

We expect that either dogs can find their way to the owner following their own scent trail back (a strategy called 'tracking') or they can perform true navigation, the ability to home over large distances without relying on route-based landmarks or information acquired during the displacement, a strategy we termed 'scouting', searching for a new way. While tracking may be safe, it is lengthy. Scouting enables taking shortcuts and might be faster but requires navigation capability and, because of possible errors, is risky.

Using GPS data complemented by video recordings by action cams, we analyzed orientation of free-roaming scent hounds. We expected to find evidence for either tracking or scouting, to test the predictions about the route length and duration, and to see indications for the type of decision made at the turning point. Furthermore, we expected that should visual piloting (orientation towards landmarks) take place there would be an effect of the height of the dog as higher (taller) dogs should have better overview (farther horizon) than lower dogs.

Altogether, 27 hunting dogs from ten breeds were equipped with a GPS collar and in several cases also with an action camera capturing a portion of the dog's head, thus providing information about its movement, behaviour, and the landscape in front of and around the dog (*Figure 1*, *Video 1*). In total, 622 trials (excursions) were performed at 62 locations during diurnal walks in forested hunting grounds in the Czech Republic from September 2014 to December 2017. Based on the records, we determined turning points, dividing the whole excursion into the outbound and inbound tracks, and we measured azimuths at particular points, length and speed of particular tracks (*Figure 2A*).

## Results

### Return strategy

In 399 cases (59.4 %), dogs homed by following their outbound track (tracking strategy), and in 223 cases (33.2 %), dogs homed using novel route (scouting strategy). In 50 cases (8.0 %), dogs combined both strategies during a single return (*Figure 2B–C*). In this study, we focus only on 'scouting'.

### Speed and track length

Scouting dogs were faster than tracking ones (*Figure 3*). As expected, taller dogs ran faster than smaller ones, but the shoulder height did not affect length of the inbound track (*Figure 4*) and the average speed of the inbound trajectory was faster when a portion of the return followed forest paths (*Figure 5*). Inbound track length was significantly longer when forest paths were used (*Figure 5*). Shoulder height did not affect inbound track length (*Supplementary file 1B*- Table 2). As expected, there was a positive correlation between direct ('beeline') distance between the turning point and the owner and the average inbound track length (*Figure 4*, *Supplementary file 1B*- Table 2).

### Azimuths and the 'compass run'

The compass directions in which dogs started the excursions (*Figure 2A*, azimuth A) and at the turning point relative to the excursion start, and thus to the goal (*Figure 2A*, azimuth B) were random, irrespective whether dogs were later tracking or scouting (*Figure 6*, *Supplementary file 1E*- Table 5). The direction in which the dogs started to return (*Figure 2A*, azimuth C) was random in tracking dogs, but significantly aligned along the ~north-south magnetic axis in scouting dogs (*Figure 7*, *Supplementary file 1E*- Table 5). Specifically, dogs homing by scouting started their return with a short (average length 18.1 m, *Supplementary file 1D*- Table 4), 'compass run' aligned along the ~north-south geomagnetic axis (*Supplementary file 1E-F*- Tables 5-6, *Video 2*).

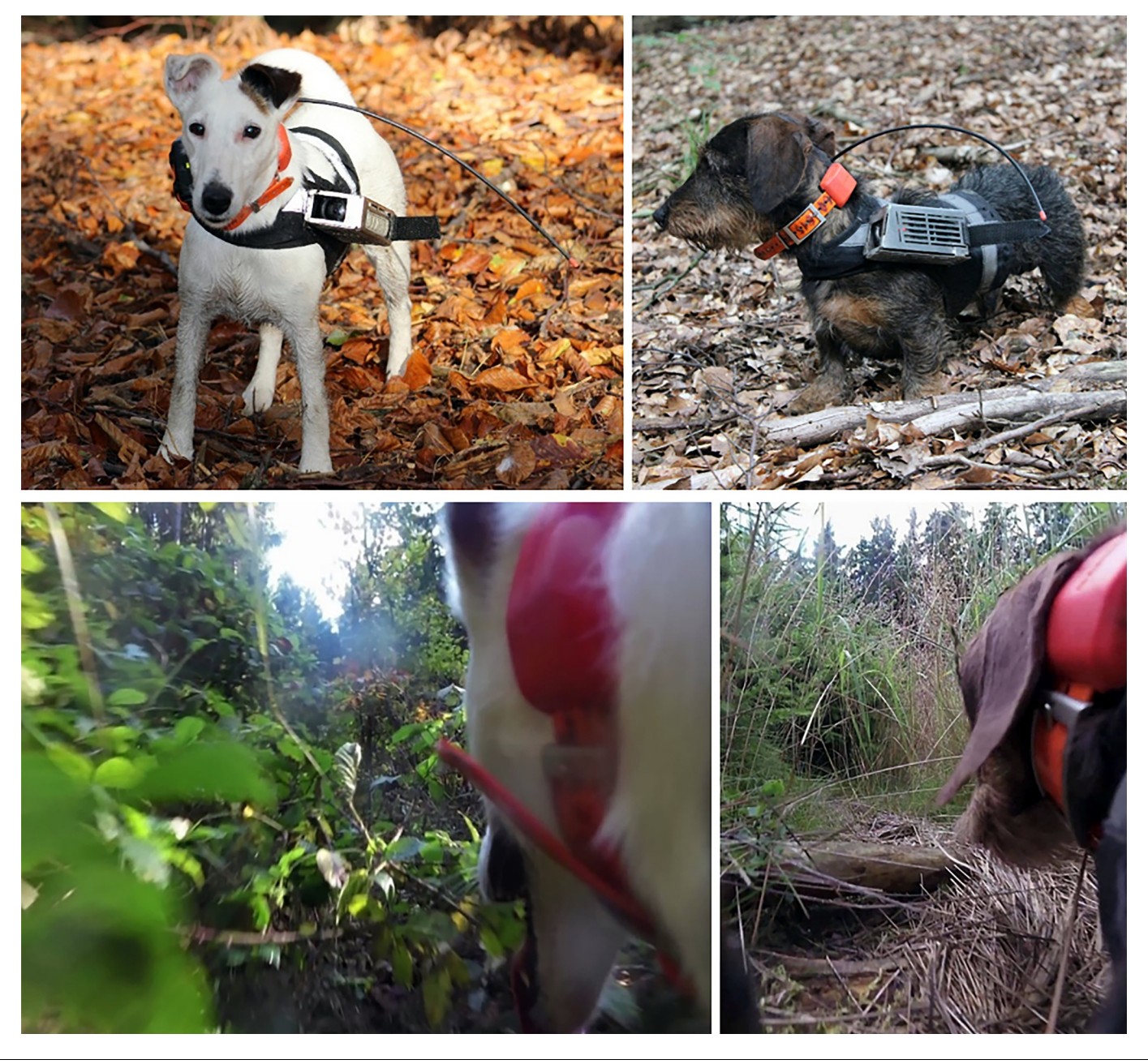

**Figure 1.** Still shots of a fox terrier (left column) and a miniature dachshund (right column) used in this study showing the tracking equipment and habitat. Above: The GPS transmitter and antenna are fixed to a collar and fitted around the animal's neck (note that for safety and comfort of the animal, the collar is free to rotate). The black fabric harness is secured around the torso and chest and is used to attach the protective camera case containing the camera and battery. Below: A typical field of view captured by the video camera that includes the dog's head/neck, and provides a detailed view of the surrounding terrain. An on-board microphone (not visible) is used to record audio.

To determine if the position of the owner influenced the orientation of the compass run, we partitioned the data into four distributions (north, south, east, or west, + / - 45°), according to the location of the owner relative to the turning point. In all four distributions, the compass run was significantly orientated along the ~north south geomagnetic axis, suggesting that its orientation was independent of the direction to the owner (*Figure 8*, *Supplementary file 1F*- Table 6).

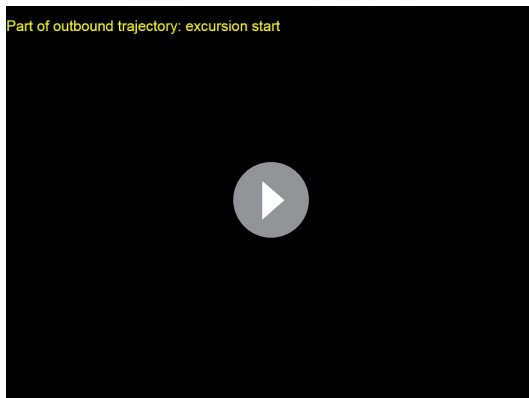

**Video 1.** Example of all three phases of an excursion. Labels of the left side of the video indicate the segment of the excursion. The video begins with the excursion start (i.e. the beginning of the outbound trajectory) when the dog becomes separated by >100 m from the owner, at which point the owner remains stationary in the forest. Shortly after, the dog detects and follows the olfactory track of a wild game animal, indicated by barking behaviour. After the outbound trajectory, the dog begins the turning trajectory phase of the excursion (see Materials and methods) and the turning point is shown when the dog briefly pauses. This location marks the beginning of the compass run (=alignment run, azimuth C, initial inbound segment). Lastly, the inbound trajectory is shown representing the phase when the dog is homing back to the location of the excursion start/owner.

https://elifesciences.org/articles/55080#video1

There were no significant differences in axial preference of the compass run between sexes (Watson's $U^2$ test, U = 0.027, p>0.5) or between familiar and unfamiliar areas (Watson's $U^2$ test, U = 0.036, p>0.5).

The probability of exhibiting a scouting strategy after compass run was aligned along the north-south axis was almost four times higher than the probability of exhibiting tracking (odds ratio = 3.60, p<0.0001) (*Figure 7E*, *Supplementary file 1B*- Table 2). No other factors appeared influential.

Importantly, when the compass run was aligned along the ~north-south axis, homing was more efficient, i.e., the ratio between the length of the inbound track and the shortest distance between the turning point and the goal was significantly reduced compared to the ~east-west compass runs (F = 6.47, p=0.01) (*Figure 9* *Supplementary file 1B*- Table 2).

## Effect of sex, breed and study site familiarity

We partitioned the data by sex for all individuals and used a Rayleigh test to determine if sex influence the orientation of the 'compass run' (*Figure 2*, azimuth C) during scouting returns. There was no significant difference between the two resulting distributions (Watson's U2 test, U = 0.027, p>0.5), and therefore, no sex differences in the orientation of the compass run.

Sex and breed did not influence the probability of return strategy used during homing (GLMM, p>0.05). Furthermore, we tested whether study site familiarity influenced the orientation of the compass run. We grouped azimuth C into two groups, according to whether the dog had visited the study at least one time prior to the excursion or if the dog was experiencing the study site for the first time. No differences between the two distributions (familiar vs unfamiliar) were found (Watson's U2 test, U = 0.036, p>0.5).

## Effect of sun

To test for an effect of sun and/or polarized light on the orientation of the compass run during scouting returns, the sun position was determined by estimating the sun's azimuth during the 15th day of each month, therefore accounting for seasonal variation in azimuth position (sun position data was taken from a central location in the Czech Republic, central to the locations of all test sites). Next, the sun azimuth direction was recorded for each hour during the 15th day of each month, for all available daylight hours. Thus, we created an average sun azimuth direction for each month of the year, with one-hour resolution. For circular analyses, we pooled the orientation of azimuth C relative to the sun position for each excursion, using the nearest hour of sun position according to each excursion time. A Rayleigh test was used to determine if the distribution of azimuth C was non-random when plotted relative to sun position. The position of sun, and thus polarization pattern, did not significantly influence the orientation of azimuth C during scouting returns at the individual level (n = 251, μ = 69°/249°, r = 0.04, p=0.673) or at the group level (n = 27, μ = 146°/326°, r = 0.021, p=0.989).

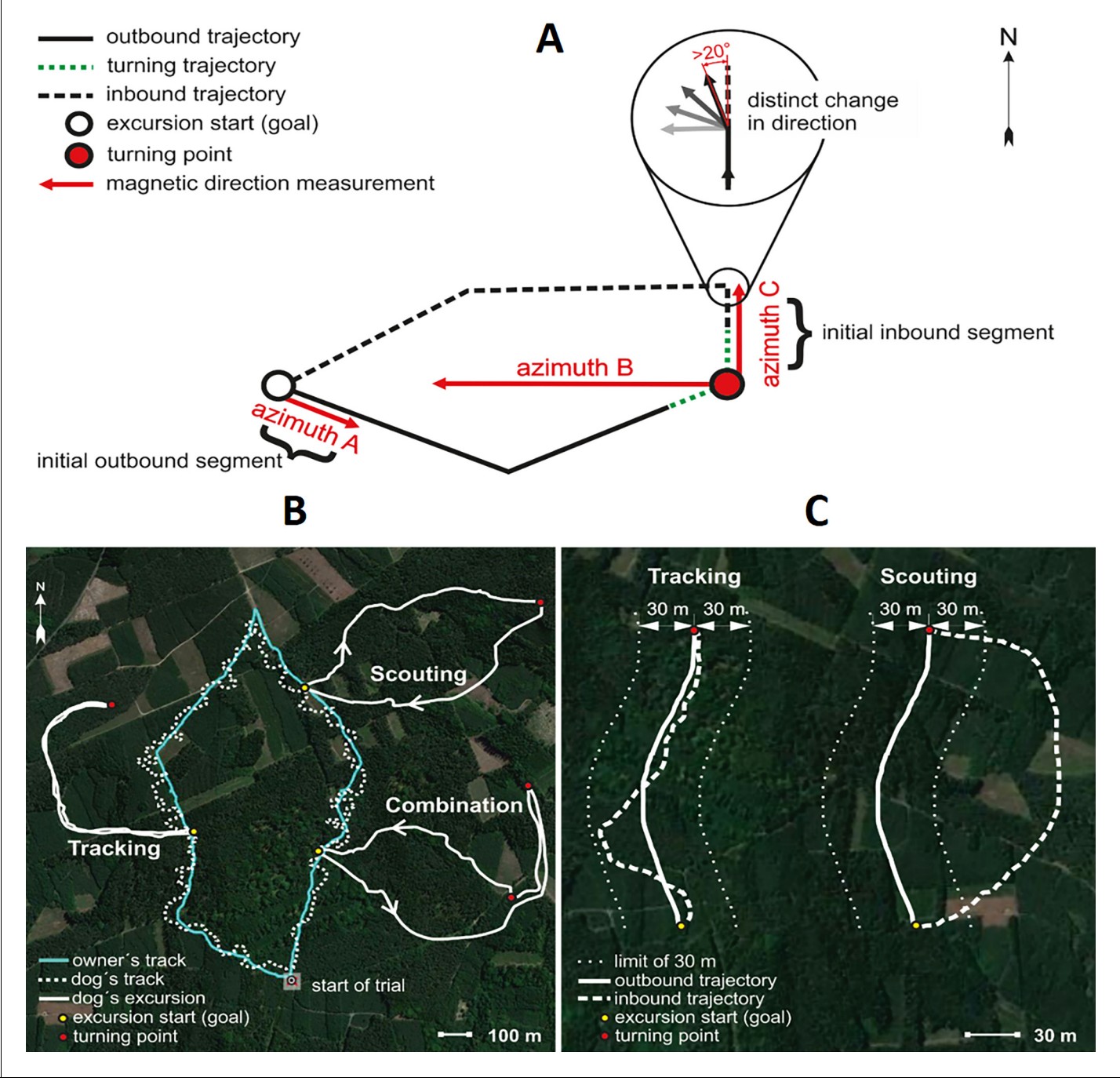

**Figure 2.** Spatial features and return strategies derived from GPS data used in analyses. (A) Schematic illustration of total excursion track. Excursion start marks location of owner when the dog is more than 100 m away, indicating that the dog is pursuing a game animal. Excursion start also marks the approximate location where the owner remains hidden until the dog returns. Turning point represents the location where the dog initiated its return to the owner. Azimuth A represents the magnetic direction of the initial outbound segment, calculated between the excursion start and the GPS point recorded 5 s after the excursion start. Azimuth B represents the magnetic direction of the owner relative to the dog at the turning point. Azimuth C represents the magnetic direction of the initial inbound segment, the 'compass run', calculated by measuring the direction between the turning point and the point where the dog exhibits a distinct (>20°) deflection in track direction (magnified inset). See Materials and methods for additional details. (B) GPS tracks showing examples of Tracking and Scouting strategies or combination of strategies. Solid white tracks show excursions. The turquoise line shows the owner's track and the white dotted line shows the dog's track during non-excursion portions of the trial. (C) Examples of Tracking and Scouting return strategies: Tracking: inbound return track falls within a ± 30 m corridor limit (shown by small white dots flanking each side of the track) of the outbound track. Scouting: the inbound return track is separated from the outbound trajectory by at least 30 m.

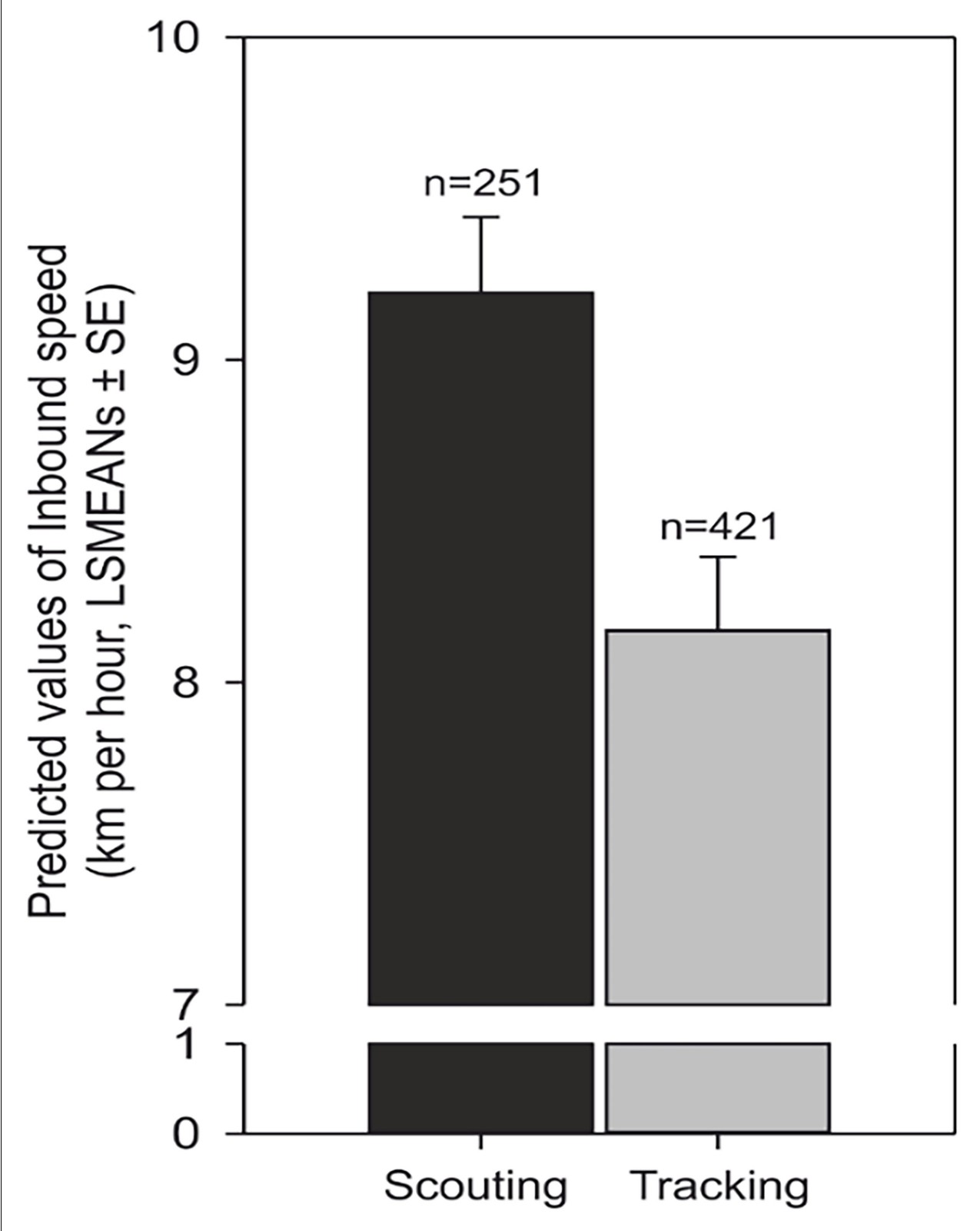

**Figure 3.** Average speed increases in scouting compared to tracking. Predicted values of inbound speed (km/hour, LSMEANs ± SE) according to return strategy and independent of the direction of the compass run (azimuth C).

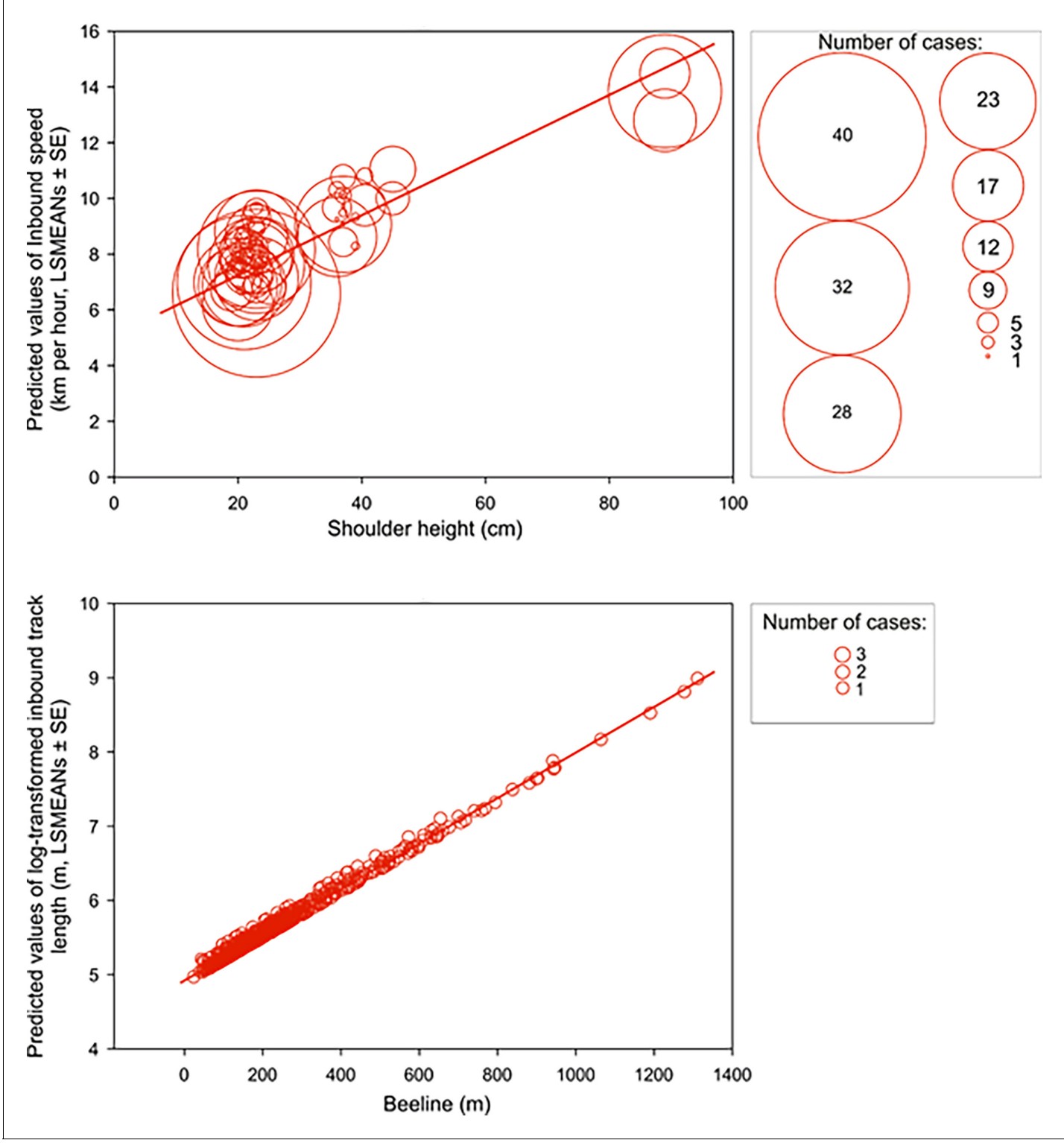

**Figure 4.** Inbound speed and track length positively correlate with shoulder height and beeline excursion distance, respectively. Upper row: A bubble-plot of predicted values of inbound speed (km per hour, LSMEANs ± SE) plotted relative to shoulder height (cm). The center of each bubble represents the predictive value and bubble size represents the number of cases for that value. Size class scale is shown on the right. Bottom row: A bubble-plot of predicted values of log-transformed total inbound track length (m, LSMEANs ± SE) as a function of direct ('beeline') distance between the turning point and the owner. The center of each bubble indicates the predictive value and bubble size is equivalent to the number of cases, as shown in the box on the right.

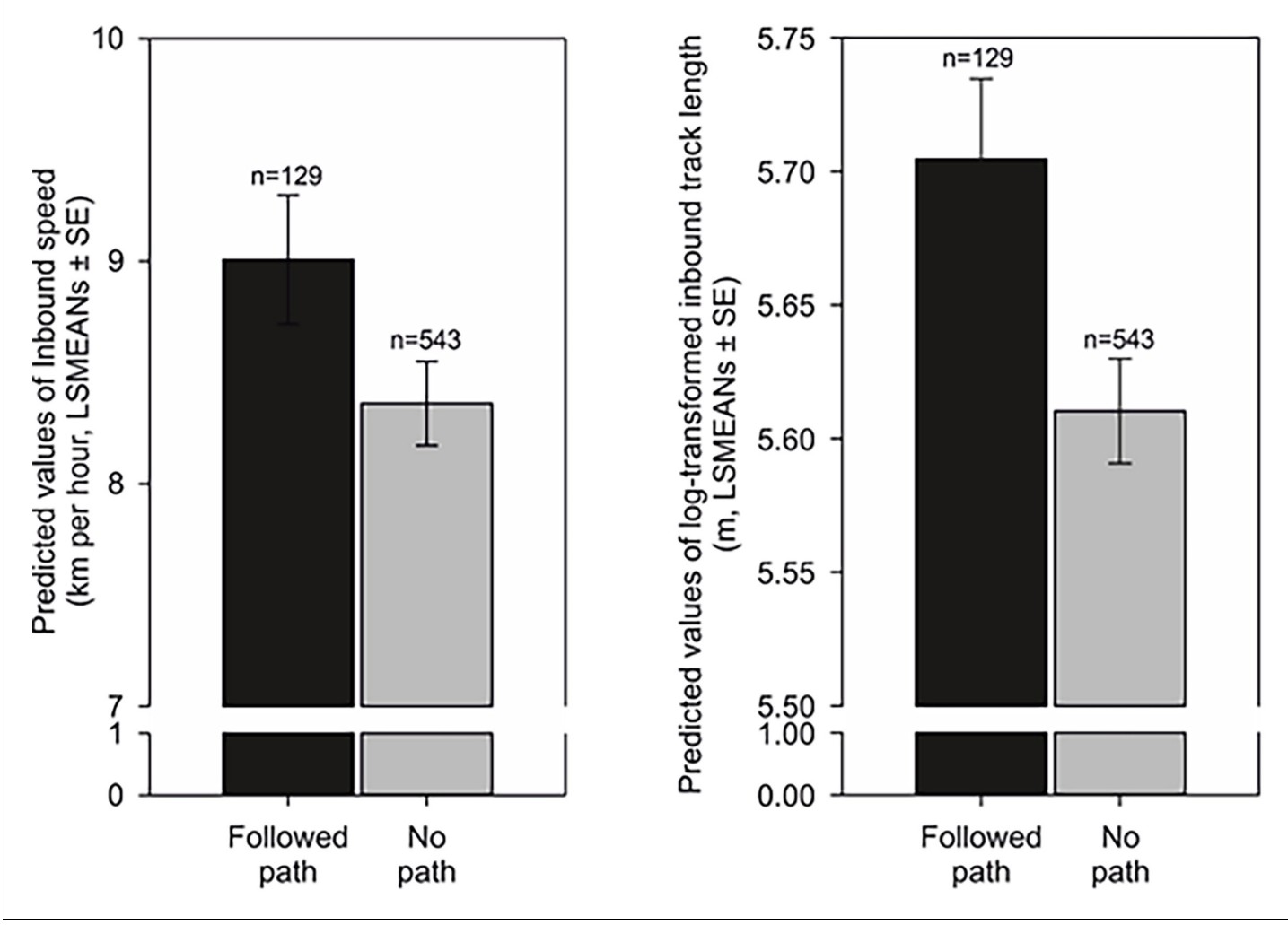

**Figure 5.** Inbound speed (length) and inbound track length (right) influenced by forest paths during the homing return. Left: Predicted values of inbound speed (km per hour, LSMEANS ± SE) grouped according to whether a portion of the inbound trajectory followed a forest path ('Followed path'), or if the return was completed without the use of a forest path ('No path'). Right: Predicted values of log-transformed total inbound track length (m, LSMEANS ± SE) between returns when at least a portion of the inbound track followed a forest path compared to returns when no forest paths were used.

## Effect of wind

Wind direction was recorded at the study site for each excursion by the owner. To test for an effect of wind direction, particularly in conditions when wind was blowing from the owner to the direction of the dog at the turning point, we used a Rayleigh test to determine if the wind direction was non-random across excursions when dogs used a scouting return strategy. In 55 cases (22 % of all scouting excursions), no wind was detected and therefore these excursions could not be included in the analyses. For the remainder of excursions (n = 196), Azimuth B (i.e. direction between the turning point and owner) (*Figure 2*) was plotted relative to the wind direction.

Partitioning the circular data into eight equal bin sizes (each bin ±22.5° and centered on the 45's, i.e. 0°, 45°, 90°, 135°, etc.), we found that the wind conditions in only 24 scouting returns (<10 % of scouting returns) were suitable for olfactory piloting to the owner (i.e. <10 % of scouting returns had wind conditions where the wind was blowing in the direction (±22.5°) from the owner to the dog at the point of return/start of azimuth C.

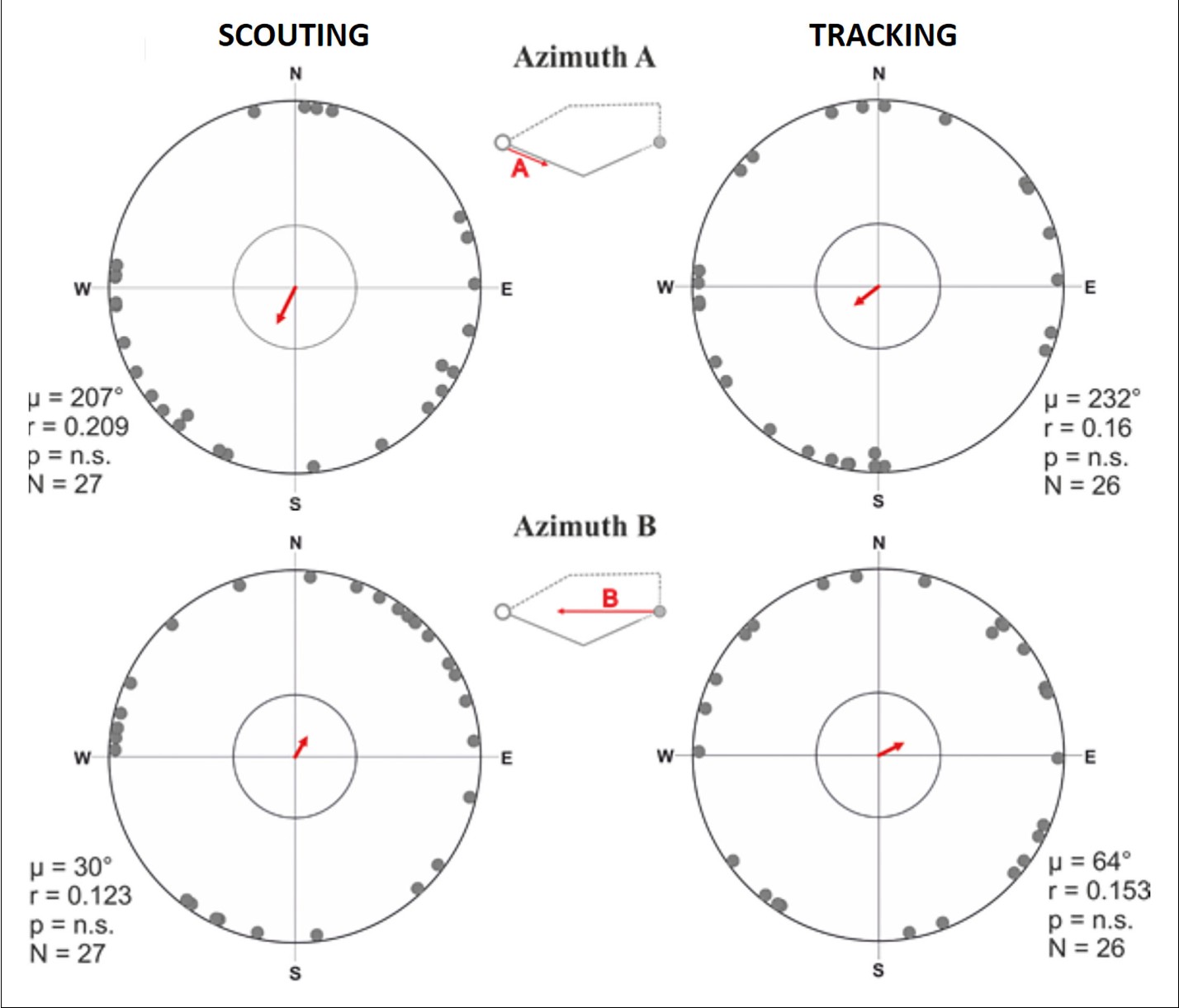

**Figure 6.** Circular distributions for azimuth A and azimuth B means grouped by return strategy. Circular distributions of magnetic orientation of the direction of the turning point relative to the excursion start/owner (azimuth B) and the initial outbound segment (azimuth A) for scouting (left column) and tracking (right column). The small schematics centered between each plot show the vector corresponding to the data in each distribution. The red arrow indicates the angular vector (μ) calculated over all angular means. The length of the mean vector (r) corresponds to the degree of clustering in the distribution on a scale of 0.0–1.0, where the circular plot radius = 1.0. The inner circle marks the 0.05 level of significance limit computed using the Rayleigh test.

## Discussion

We found that dogs returning in a forest either follow back their outbound trajectory, a strategy called *tracking* or chose a completely new route, a strategy called *scouting*. In this study, we analyzed only scouting events and found a conspicuous phenomenon. In most cases, dogs start their return with a short (about 20 m long) run, called here *compass run*, mostly performed along the north-south axis irrespective of the actual homeward direction.

It is unlikely that the direct involvement of visual, olfactory or celestial cues can explain the highly stereotyped and consistent ~north south alignment of the compass run. For example, the forested habitat and dense vegetation of the study sites make visual piloting unreliable and, in many cases,

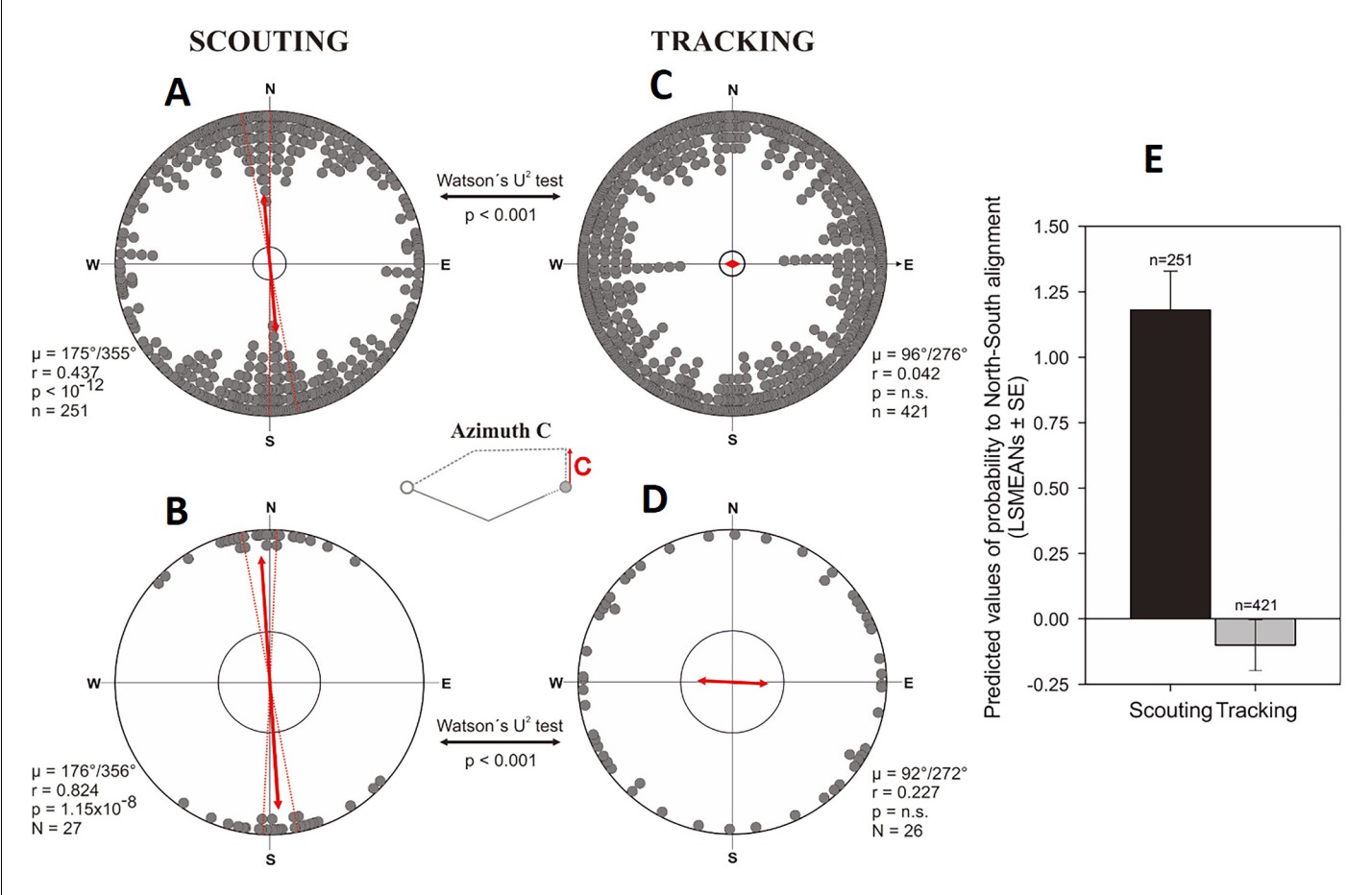

**Figure 7.** Alignment responses during the initial inbound return (= 'compass run') in free-roaming dogs. (A–D) Circular distributions showing geomagnetic alignment responses during the initial inbound segment (azimuth C, 'compass run'), when distributions are partitioned into Scouting (A, B) and Tracking (C, D) return strategies. Grey bearings plotted on the periphery of the distributions represent the axial orientation of compass runs for each excursion (A, C), each bearing treated as an independent data point, or (B, D) the axial orientation of compass runs when the mean orientation was first calculated for each dog. All data are plotted relative to magnetic north, indicated at the top of each plot, and the red double-headed arrow indicates the mean axial vector (μ) for each distribution. The length of the mean vector (r) corresponds to the degree of clustering around the mean and ranges from 0.0 to 1.0, with the radius of each plot = 1.0. Dashed red lines represent the 95% confidence intervals and the inner grey circle marks the p=0.05 level of significance limit computed using the Rayleigh test. Results from Watson's $U^2$ tests are shown between distributions, revealing significant differences in the compass run orientation between tracking a scouting return strategies. The small schematic centered between the plots shows azimuth C and the axial direction of the red vector corresponds to the orientation data plotted in each distribution. (E) Predicted values of the probability that dogs will exhibit a compass run along the ~north south geomagnetic axis (±45°) during the initial inbound segment (LSMEANs ± SE) according to return strategy.

not possible. Furthermore, there was no effect of the body height (and thus the degree to which the dog's field of view of its surroundings was limited) on the probability of north-south alignment when compared to east-west alignment, a fact which is not consistent with a visual piloting hypothesis. Highly variable wind conditions, coupled with turbulence in the forest understory, rule out the use of olfactory piloting during scouting. In many cases, the Sun's disk was fully obstructed by cloud cover and/or overhead vegetation, making it challenging to use a sun or polarized light compass. And, although polarized light has been shown to calibrate the magnetic compass in bats (*Greif et al., 2014*), detection of skylight polarization is not thought to be widespread in mammals (*Horváth and Varjú, 2004*; *Marshall and Cronin, 2011*). More generally, it is difficult to reconcile what advantage a north-south orientation response would provide for any of the sensory modalities discussed above.

In contrast, the Earth's magnetic field provides a stable, omnipresent cue, regardless of daily or seasonal temporal variation, visual cue availability or weather conditions. The north-south alignment of the compass run in dogs is consistent with a wealth of studies providing support for spontaneous

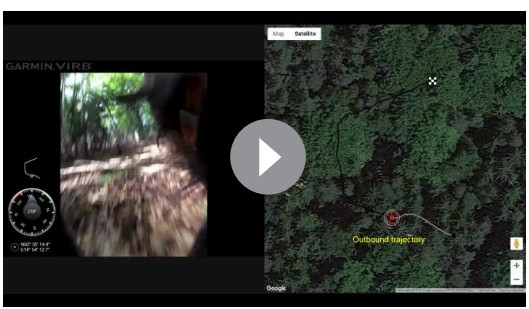

**Video 2.** Example video showing the compass run behaviour during a scouting strategy return. The GPS track is shown on the right half of the video with the red 'bulls-eye' tracker synced with the video shown on the left. The bulls-eye tracker moves across the track corresponding to the position of the dog in the video. The checkerboard square represents the location where the video ends, but does not represent any relevant features/landmarks used for analyses. Compass and GPS measurements are shown on the left and are synced to the video and bulls-eye tracker. Yellow labels appear superimposed on the right side of the screen to indicate relevant features of the excursion. Before reaching the turning point, the dog is travelling in a ~ west northwest direction. As shown, the compass run (=alignment run, azimuth C) starts at the turning point and the dog begins its initial inbound segment in the ~north direction. The compass run is terminated at 0:22 when the dog changes magnetic direction by >20° (see Materials and methods) and the inbound trajectory continues as the dog navigates back to the owner (not shown).

https://elifesciences.org/articles/55080#video2

magnetic alignment along the north-south magnetic axis in a range of vertebrates in the field (reviewed in *Begall et al., 2013*; *Malkemper et al., 2016*) as well as in the laboratory under controlled conditions (e.g. *Burda et al., 1990*; *Phillips et al., 2002*; *Muheim et al., 2006*; *Malkemper et al., 2015*; *Painter et al., 2018*).

While the functional significance of magnetic alignment is not fully understood, magnetic alignment may help to organize and structure many aspects of spatial behaviour (*Begall et al., 2013*). This may help to explain the compass run at the beginning of homing in scouting dogs. Indeed, several recent studies have shown that dogs are sensitive to magnetic cues (*Hart et al., 2013*; *Adámková et al., 2017*; *Martini et al., 2018*), and behavioural studies show that Earth-strength magnetic fields play a direct role in homing responses also in the blind mole-rat, a subterranean mammal (*Kimchi et al., 2004*). Taken together, we propose that the compass run is mediated by magnetic cues, and it helps to increase the accuracy and reduce the complexity of long-distance navigation across unfamiliar and/ or highly variable environments.

Noteworthy, scouting dogs in unfamiliar locations cannot use visual landmarks to recalibrate a path integration system. Therefore, in the absence of familiar landmarks, the compass run may serve to recalibrate a path integration system relative to Earth's magnetic field, so that errors accumulated during the outbound trajectory are not incorporated into the estimate of the homing direction. Importantly, dogs in our study were not passively displaced as is usual in most homing experiments (*Tsoar et al., 2011*; *Ostfeld and Manson, 1996*), and therefore, the involvement of path integration seems plausible, and may be one of several reasons why the compass run has not been identified in previous studies.

Our findings clearly show the importance of further research on the role and involvement of magnetic cues in canine (and more generally mammalian) navigation. More specifically, the research suggests that the magnetic field may provide dogs (and mammals generally) with a 'universal' reference frame, which is essential for long-distance navigation and arguably the most important component that is 'missing' from our current understanding of mammalian spatial behaviour and cognition.

## Materials and methods

### Study subjects

A total of 27 hunting dogs (10 M, 17 F) from ten breeds were used in the study (*Supplementary file 1A*- Table 1). All dogs come from breeds with pedigrees in hunting and animal tracking and were regularly assessed by veterinarians throughout the study. These dogs innately detect and pursue the olfactory tracks of game, and in rare cases, dogs were able to spot game animals from a distance (i. e. >20 m) through the forest. However, the small breeds used in this study are unable to keep pace with the much larger and faster game animals. Therefore, dogs were almost immediately left to rely on olfactory tracking and never posed any physical threat to wild animals.

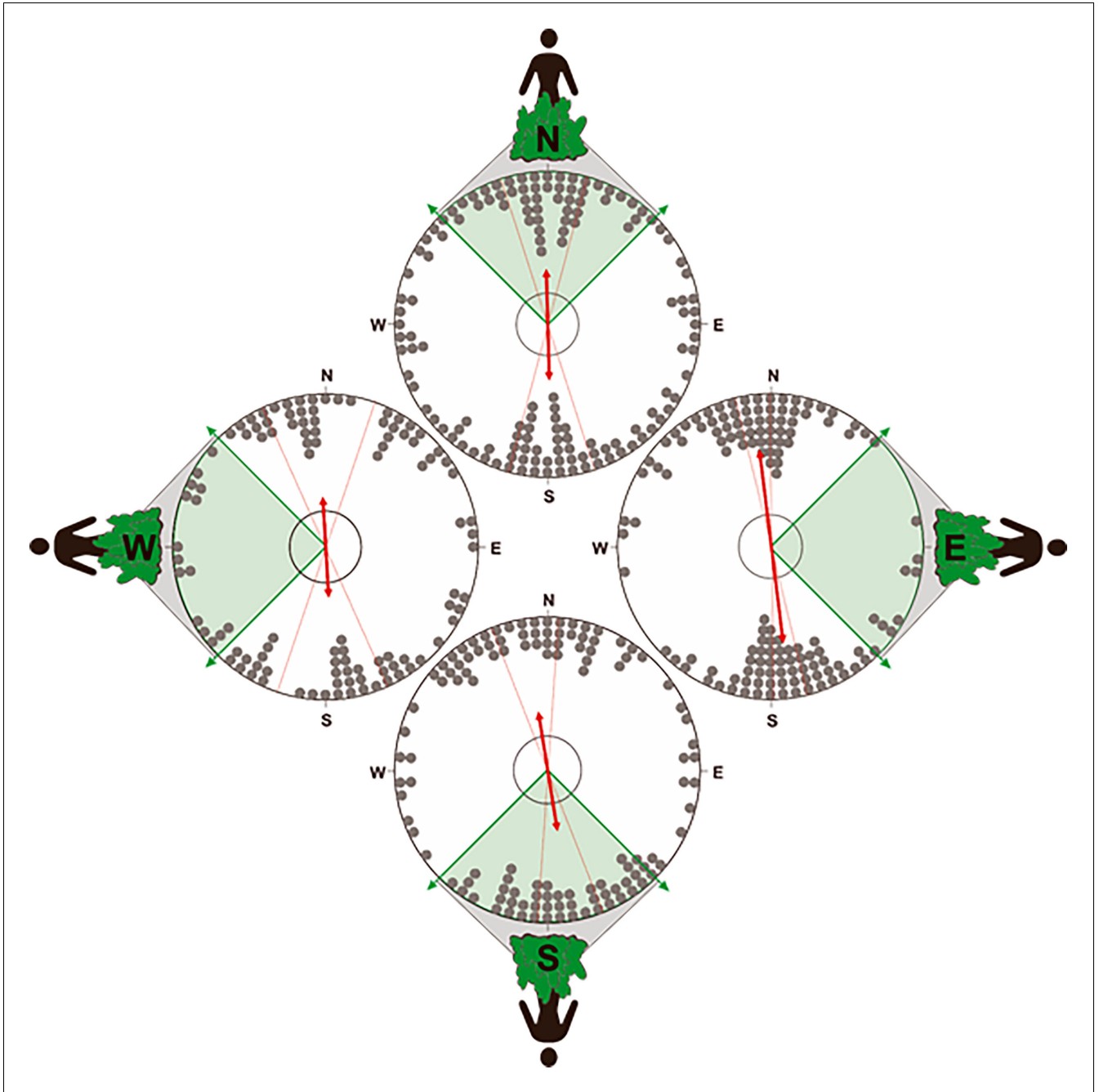

**Figure 8.** Orientation of the compass run plotted relative to the position of owner. To test for an influence of the owner on the orientation of the compass run (azimuth C) during scouting, the data was partitioned into four distributions corresponding to when the owner was located at one of the four cardinal compass directions (±45°) relative to the turning point. Therefore, the distributions above show the orientation of azimuth C when the owner is positioned at magnetic ~north (316°−45°),~east (46°−135°),~south (136°−225°), or ~west (226°−315°) relative to the turning point (i.e. relative to the start of the compass run). If the N-S orientation of the compass run is a direct response to the owner (i.e. olfactory or visual piloting towards the owner), then in situations when the owner is located along the ~east west axis relative to the turning point, the orientation of the compass run should also align along the ~east-west axis. Alternatively, if the compass run is independent of the owner, then there should be no statistical difference in the orientation of azimuth C when the owner is position in different magnetic directions. Each distribution plots the axial bearings recorded from each excursion. The position of the owner relative to azimuth C is shown on the outside of each distribution and the shaded section of each plot bounded with green arrows shows the range of possible positions of the owner in each distribution (45° range centered on each cardinal compass direction). All other symbols are identical to those in *Figure 6*. Note that the axial bearings have been rounded to the nearest 5° only to facilitate the graphical representation of the responses and all statistics were calculated using data values measured to 1° resolution.

## Tracking equipment

Dogs were equipped with a Garmin T5 mini (or DC50) GPS collar (Garmin Ltd., USA) fitted around the neck and programed to record GPS positions at 2.5 or 5.0 s intervals (GPS accuracy ±4.2 m based on ground-truth measurements collected at study site locations). A portion of the trials (31 %) were carried out using a Garmin Virb Elite action camera (Garmin Ltd., USA) housed inside a non-magnetic stainless steel mount fixed to the side of a custom-fitted fabric harness (*Figure 1*). The camera captured a similar field of view to that of each dog (*Figure 1*), and thus provided information about the dog's visual surroundings, including habitat characteristics. In addition, a portion of the dog's head was also captured, providing information about the dog's behaviour and movement (e.g. activity, head scanning, head orientation) and an on-board microphone detected barking behaviour, an indicator that the dog was in pursuit of a game track (*Video 1*). The customized harness was designed to minimize discomfort and allowed full mobility for all dogs. Dog owners were equipped with a handheld Garmin Alpha 100 or Astro 320 receiver (Garmin Ltd., USA) used to monitor and record GPS tracks.

## Experimental procedure

A total of 622 trials were performed in forested hunting grounds at 62 independent locations in the Czech Republic from September 2014 to December 2017. All trials were performed with a single dog (i.e. trials were not carried out with groups of dogs), in areas free from high voltage power lines, paved roads or buildings, and at different times of day (daylight hours only) and different times of year. Dogs were transported to each site by a car and were given a 10 min rest and acclimation period in the close surroundings of the car (<20 m radius). Dogs were then equipped with the GPS collar, and in some cases, the harness-camera setup, and walked off-leash alongside the owner into the surrounding forest. Dogs could freely roam and explore the area to search for wild game tracks using olfactory cues, as is an innate behaviour in this context for the breeds used in this study. The following game animals are common in the study region: fallow deer (*Dama dama*), red deer (*Cervus elaphus*), roe deer (*Capreolus capreolus*), wild boar (*Sus scrofa*), European hare (*Lepus europaeus*), and red fox (*Vulpes vulpes*). During the search period, owners did not provide visual or acoustic commands to instruct the dog. The handheld GPS device was programmed to indicate when the dog had travelled ≥100 m from the position of the owner. At this moment (designated as 'excursion start') the owners stopped walking and marked their location on the handheld GPS. Owners then hid behind trees or dense vegetation within a 10 m radius from the 'excursion start' to minimize the possibility of visual piloting by dogs in final stages of their inbound return (see below). The owners remained at this place until the dog returned. The location of each trial, the dog's familiarity with the location, and weather conditions (wind speed, wind direction, and temperature) were recorded. Locations visited for the first time were considered to be unfamiliar, whereas dogs who had visited the location at least once previously were considered familiar with the area. The entire trial, including excursions (see below), lasted between 30 and 90 min.

## Analysis of excursions

### Identifying excursion features

'Excursions' are defined as the track recorded between the excursion start (see above) and the point when the dog started its return to the owner, staying at, or within 10 m from the excursion start (*Figure 2*). All excursions were analyzed using Garmin BaseCamp 4.6.2 software (Garmin Ltd., USA). In very rare cases when dogs did not return to the owner, they were found via GPS position and excluded from further analysis (n = 16, 2.4 % of all excursions).

Based on preliminary analyses of GPS tracks used to standardize the study protocol, excursions were divided into three distinct phases: 1) *outbound trajectory*: from the excursion start to the point where the dog terminates its pursuit of the game track; 2) *turning trajectory*: the part where the dog initiates its return to the excursion start/owner; within this part, we narrowed the *turning point*; 3) *inbound trajectory*: the return track to the owner (*Figure 2A*). We expected that turning trajectory (and specifically turning point) would be characterized by slowing down, perhaps even short stop as the dog orientates.

The whole excursion length (=100 %) was divided into ten equidistant segments (i.e. each segment encompassing 10 % of the total excursion length). The average speed over each segment was

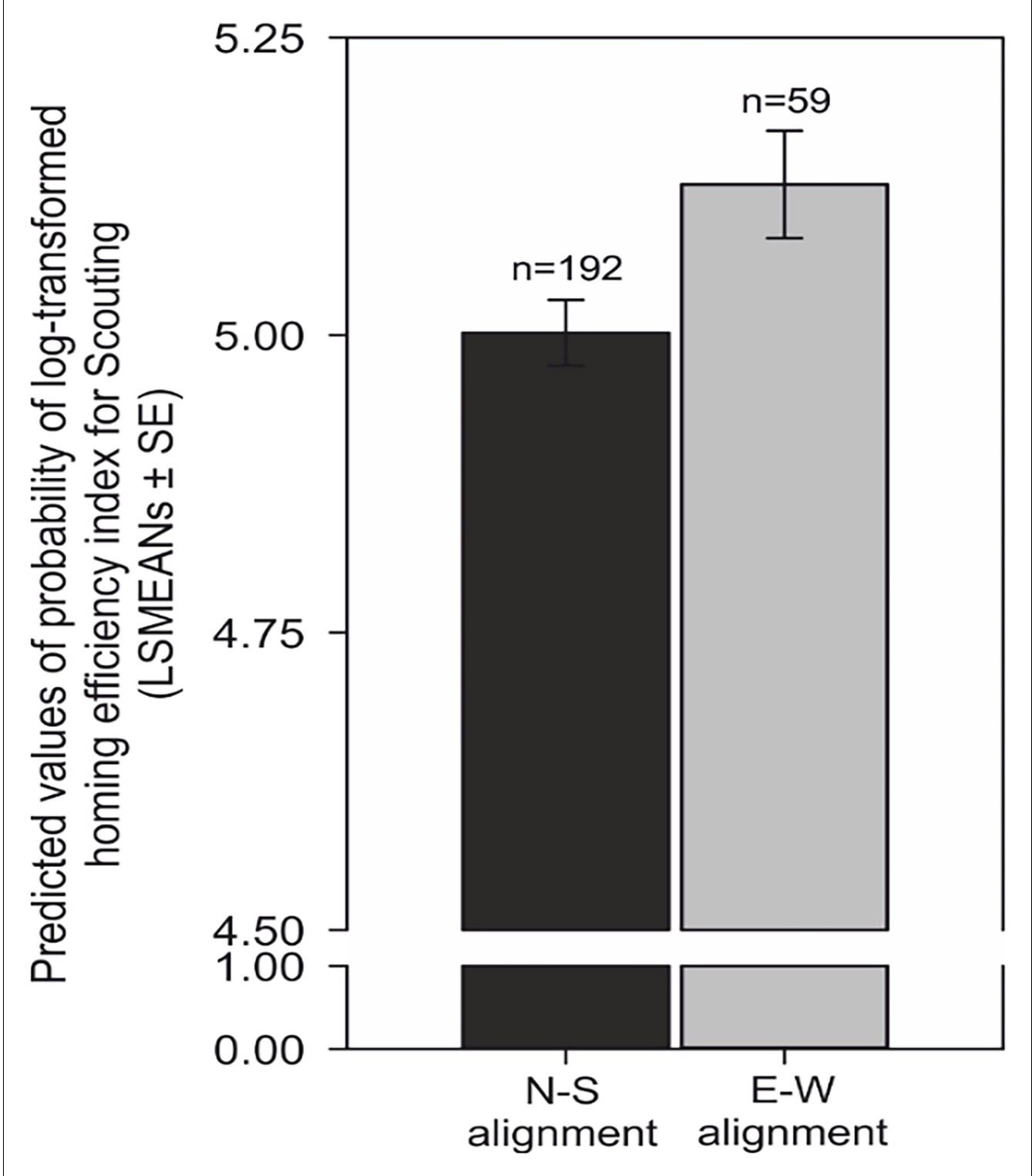

**Figure 9.** Compass run improves homing performance during scouting. Comparison of predicted values of the probability of log-transformed homing efficiency index (LSMEANS ± SE) between dogs exhibiting a compass run oriented along the ~north south (±45°) compared to orientation along the east-west (±45°) axis during scouting return strategies. The efficiency of homing is expressed as the ratio (%) between the length of the dog's inbound trajectory and the direct (beeline) distance between the turning point and the excursion start/owner. Titles and legends to supplementary videos.

calculated and the segment with the slowest speed was labelled as the *turning trajectory*. If the so determined place seems to be improbable (either too close to the start/goal or the dog pauses relatively long at one point) we checked the situation at video to exclude the possibility that the dog stopped to drink, bath, sniff at some interesting place or encountered difficult terrain. If this was the case or video was not available, the second slowest segment was identified as the turning trajectory (n = 26,<5 % of all excursions). If this segment was also located near the excursion start/owner, the respective situation was again checked at the video if available, and/or the slower segment with the farthest straight-line distance from the excursion start was labelled as the turning trajectory (n = 53, 8.5 % of all excursions).

Within the turning trajectory, the average speed was calculated between each successive GPS point. The point-to-point path within the turning trajectory with the slowest speed was identified and marked as the *turning point*, representing the specific location where the dog initiated its return to the owner (*Figure 2A*, *Video 2*).

The focus of the current study was to evaluate long-distance navigation in free-roaming dogs, and therefore, excursions shorter than 200 m were excluded from the analyses.

## Identifying azimuths

All magnetic measurements used in the analyses were made using measurement tools in Garmin BaseCamp 4.6.2 software and magnetic declination was taken into account. The magnetic direction between the point of the excursion start and the GPS point recorded five seconds after the excursion start was measured (i.e. the initial outbound segment) and is defined here as azimuth A (*Figure 2A*). Azimuth B represents the magnetic direction of the owner relative to the dog at the turning point (*Figure 2A*). The magnetic direction of the initial inbound segment, azimuth C (denoted here as *compass run*), was determined by measuring the direction between the turning point and the point where the dog exhibited a > 20° deflection in track direction without an immediate return to its preceding track heading (*Figure 2A*). These criteria helped to omit short-lived track deviations often caused by obstacles (e.g. fallen trees, dense clusters of vegetation) and were applied to all tracks. A criterion for classifying the compass run as either north-south or east-west, was established by grouping runs into one of four sectors corresponding to a sector of ±45° of the cardinal compass axes, i.e. ∼north (316°−45°),∼east (46°−135°),∼south (136°−225°), or ∼west (226°−315°).

## Return strategies

Based on preliminary evaluations from a subset of excursions, each inbound return was categorized into two distinct return strategies:

1. Tracking: The inbound return trajectory followed the outbound trajectory, i.e. the dog 'simply' followed its outbound track back to the owner (*Figure 2*). Here, the inbound return track is no more than 30 m from the outbound track at any point along the return path (*Figure 2*).
2. Scouting: A novel route of return was taken to the owner (i.e. dog was not following its outbound trajectory) (*Figure 2*). Here, the inbound and outbound trajectories were separated by more than 30 m (*Figure 2*).

In some cases, dogs exhibited a combination of return strategies, e.g. dogs began the inbound return trajectory using a tracking strategy and later changed to a scouting strategy, or vice versa (*Figure 2*). In these relatively rare situations (n = 50, 8.4 % of all excursions), strategies were divided into two separate tracks and the initial inbound segments (azimuth C) were measured for each strategy.

Importantly, the personnel responsible for identifying the spatial features of excursions (i.e. excursion start, outbound, turning and inbound trajectory, turning point and owner position, see Identifying excursion features) as well as partitioning tracks into return strategy type were unaware of the directional data (azimuths A, B, C). Conversely, personnel responsible for measuring directional data were unaware to which segments and strategies each measurement belonged. Therefore, the analysis of all directional data was carried out using a double-blind protocol.

## Statistical analysis

Circular statistical analyses were carried out with Oriana 4.02 (Kovach Computing Services). Before evaluation, all directional responses were grouped by return strategy. Magnetic headings for azimuth A and azimuth B were treated as angular data. However, preliminary results for azimuth C revealed a strong bimodal response within individuals, and therefore, azimuth C was treated as axial data (*Batschelet, 1981*). The Rayleigh test was used for circular statistics to determine if distributions were indistinguishable from random at the p<0.05 significance level. To test for non-random orientation, all responses were evaluated at the individual level (i.e. measurements from excursion treated as an independent bearing) and at the group level by calculating mean directional response from each dog then calculating a grand mean vector. A Watson's U2 test was used to test for differences between distributions (*Batschelet, 1981*).

Non-circular data were analyzed using SAS System (version 9.4). Associations between inbound speed and return strategy, as well as homing efficiency and geomagnetic alignment (N-S or E-W) were tested using a multivariate General Linear Mixed Model (GLMM, PROC MIXED), with inbound speed or homing efficiency as a dependent variable. A homing efficiency index was expressed by calculating the ratio (in %) between the track length of the inbound trajectory (i.e. track length of dog between turning point and excursion start) and the direct distance (i.e. beeline) between the turning point and excursion start. Additional analyses (i.e. alignment or return strategies as dependent variables) were performed using a GLMM with PROC GLIMMIX for binary distributions. Link function was logit and the distribution of error terms was binomial in the GLMM. Since models with PROC GLIMMIX did not converge due to an effect with considerably larger classes (dog ID and/or breed), we applied the procedure with Method = Laplace (*Kiernan et al., 2012*).

Models were constructed by entering the predicted effects, i.e. return strategy for dependent variables, inbound speed and alignment (modelling the probability that geomagnetic alignment = N s); and dependent variable for homing efficiency index. Each model was checked with additional factors that could affect the model predictions (see *Supplementary file 1C*- Table 3 for list of factors). Factors which did not contribute (i.e. factor significance p>0.05) were dropped from the model. Interaction terms were tested and all tests were treated as two-tailed distributions.

The effects used in the analyses were continuous variables and classes are listed in *Supplementary file 1. C* - Table 3. Where appropriate, variables were log-transformed to improve normality of residuals and to reduce skewness. All fitted models included the dog's identity as a random effect to account for the use of repeated measures across the same individuals.

Least-squares means (LSMEAN) were computed for each class and differences between classes were tested using a t-test. Associations between the dependent variable and time were estimated by fitting a random coefficient model using PROC MIXED (*Tao and Littell, 2002*). Predicted values of the dependent variable were calculated and plotted against the fixed effect with predicted regressions for each group. Where more than one value was plotted in the same position, a bubble plot was generated to represent the data.

To compare the probability of an event between two groups, an odds ratio was calculated (*Stokes et al., 2012*). Odds ratios greater than one imply that the event is more likely to occur in the first group, while an odds ratio less than one implies that the event is more likely to occur in the second group.

## Acknowledgements

We thank the following students, colleagues and collaborators for helping to collect the field data: Tereza Březinová, Václav Fuks, Hedvika Fuksová, Mirka Jakšlová, Barbora Kletečková, Hana Kneřová, Alena Mottlová, Richard Policht, Jaroslav Spal, Hana Spalová, and Miloslav Zikmund. We thank Richard Holland, Diethard Tautz, and two anonymous reviewers for their constructive comments on the manuscript.

## Additional information

### Funding

| Funder | Grant reference number | Author |
|---|---|---|
| European Social Fund | Operational Programme Research, Development and Education, EVA 4.0 | Hynek Burda |
| European Social Fund | Operational Programme Research, Development and Education, CZ.02.1.01/0.0/ 0.0/16_019/0000803 | Hynek Burda |
| Ministry of Agriculture of the Czech Republic | MZE-RO0718 | Luděk Bartoš |
| Czech University of Life Sciences Prague | CIGA CZU (Project No. 20174319) | Hynek Burda |
| Faculty of Forestry and Wood Sciences, Czech University of Life Sciences Prague | IGA (Project No. B07/16) | Hynek Burda |
| Grant Agency of the Czech Republic | No. 15-21840S | Hynek Burda |
| European Regional Development Fund | Operational Programme Research, Development and Education, EVA 4.0 | Hynek Burda |
| European Regional Development Fund | Operational Programme Research, Development and Education, CZ.02.1.01/0.0/ 0.0/16_019/0000803 | Hynek Burda |

The funders had no role in study design, data collection and interpretation, or the decision to submit the work for publication.

### Author contributions

Kateřina Benediktová, Conceptualization, Resources, Data curation, Formal analysis, Investigation, Visualization, Methodology, Writing - original draft; Jana Adámková, Investigation, Visualization, Methodology, Project administration; Jan Svoboda, Petra Nováková, Investigation; Michael Scott Painter, Formal analysis, Methodology, Writing - original draft, Writing - review and editing; Luděk Bartoš, Formal analysis, Visualization, Writing - original draft, Writing - review and editing; Lucie Vynikalová, Formal analysis, Visualization; Vlastimil Hart, Conceptualization, Supervision, Funding acquisition, Methodology, Writing - original draft, Project administration; John Phillips, Formal analysis, Writing - original draft, Writing - review and editing; Hynek Burda, Conceptualization, Formal analysis, Supervision, Funding acquisition, Validation, Visualization, Methodology, Writing - original draft, Project administration, Writing - review and editing

### Author ORCIDs

Kateřina Benediktová (iD) https://orcid.org/0000-0003-0130-4770
Vlastimil Hart (iD) http://orcid.org/0000-0003-4901-3817
Hynek Burda (iD) https://orcid.org/0000-0003-2618-818X

### Ethics

Animal experimentation: Permission from landowners and local game managers were obtained prior to entering each location, and searching and tracking methods were in accordance with the Czech national law and regulations for game management (§ 14 and § 15, Decree No. 244/2002, Ministry of Agriculture, Statue No. 449/2001, Game Management). The Professional Ethics Commission of the Czech University of Life Sciences in Prague has decided that according to the law and national and international rules, this study has not a character of an animal experiment and does not require a special permit.

Decision letter and Author response
Decision letter https://doi.org/10.7554/eLife.55080.sa1
Author response https://doi.org/10.7554/eLife.55080.sa2

## Additional files

### Supplementary files

• Source data 1. Basic raw data used in calculations.

• Supplementary file 1. Information on subjects studied, parameters included in the analyses, and results of circular statistics. (A) Table 1 Information about dogs used in the study. Owner = initials of owner accompanying dog during walks, Age = age or age-range during the study period, $N_{OUT}$ = number of outbound trajectories, $N_{IN}$ = total number of inbound trajectories, $N_T$ = number of inbound returns using a tracking strategy, $N_S$ = number of returns using a scouting strategy. (B) Table 2 Factors in the final GLMMs for the dependent variables (in bold). a) probability for N-S alignment (±45°) during the initial inbound segment (i.e. 'compass run'); b) probability for scouting strategy; c) efficiency of return; d) speed of inbound trajectory; e) inbound track length. (C) Table 3 Effects used in General Linear Mixed Models. (D) Table 4 Length parameters during different phases of the excursion (data from combined strategies are excluded). (E) Table 5 Circular analyses of individual ('raw') and grouped means for azimuth A, B and C during scouting and tracking strategies, and when a scouting strategy was used as the second return strategy (tracking used as a second return strategy not shown). Means were calculated by averaging directional headings for each dog, then calculating a grand mean from all individuals. Raw data were calculated by treating each azimuth as an independent bearing. Note that due to the bimodal preference found within individual dogs for azimuth C, these bearings were treated as axial data. See *Figures 6–8*. (F) Table 6 Axial analyses of azimuth C (=orientation of the compass run) partitioned into four groups to test for an influence of the owner on the orientation of the compass run during scouting strategy returns. Each analysis corresponds to the orientation of the compass run when the owner was located in one of four cardinal compass directions (±45°) relative to the turning point. Therefore, owner positions relative to the turning point are: owner = magnetic ~ north (316°−45°),~east (46°−135°),~south (136°−225°), or ~west (226°−315°). All data are treated as independent bearings. (See *Figure 8*).

• Transparent reporting form

### Data availability

The raw source data are provided as Source Data 1.

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
