## [Decision Letter]

**Acceptance summary:**

The authors report a novel efficient orientation strategy in dogs which consists of homing using an inbound route different from their outbound route. Specifically, dogs run a short way along the north-south geomagnetic axis, irrespective of the actual direction homewards. This behavior is argued to serve to bring a mental map into register with the magnetic compass and to determine its polarity.

**Decision letter after peer review:**

Thank you for submitting your article "Magnetic alignment enhances homing efficiency of hunting dogs" for consideration by *eLife*. Your article has been reviewed by three peer reviewers, and the evaluation has been overseen by a Reviewing Editor and Diethard Tautz as the Senior Editor. The following individual involved in review of your submission has agreed to reveal their identity: Richard Holland (Reviewer #1).

The reviewers have discussed the reviews with one another and the Reviewing Editor has drafted this decision to help you prepare a revised submission.

Summary:

Benediktova and colleagues present an exciting study examining the navigational capabilities and strategies of hunting dogs. Using GPS tracking, the researchers studied the paths that the dogs took away from and then back to their owner and characterized the strategy employed by the dog as either "tracking", a strategy whereby dogs follow their outbound track back to their starting point, or "scouting", a strategy whereby dogs use a novel path back to their starting point. It was observed that dogs run oriented along the north-south geomagnetic axis for a short distance following the turning point of their overall route, which was observed when dogs employed a "scouting" strategy but not a "tracking" strategy. The major conclusion presented by the authors is that dogs utilize the Earth's magnetic field as a navigational cue when dogs navigate over distances of 100s of meters. All reviewers found the study of interest and agreed that as an observational study it was well conducted. However, the reviewers noted that a number of concerns need to be addressed.

Essential revisions:

1) Identification of the "turning point" and the definition of the "compass run" require further attention. At minimum the authors should provide some a priori justification for this. We would like the authors to consider providing some form of analyses of how the result change with different speed and degree of turn.

2) The argument that the "compass run" could be used to bring a cognitive map back into register was found to be vague and unclear, and the reviewers believed that there was no evidence for this. This interpretation should be revisited or additional data provided in favour for it.

3) A discordance between the Introduction and the Discussion needs to be addressed. The Introduction seems to focus on true navigation as the explanation for scouting, but the Discussion focuses on path integration and cognitive maps as the explanation, neither of which are true navigation by the standard definitions.

---

## [Author Response]

Essential revisions:1) Identification of the "turning point" and the definition of the "compass run" require further attention. At minimum the authors should provide some a priori justification for this.

We described the procedure for determining the *turning point* in the subsection ”Identifying excursion features”. Apparently, some important aspects were not adequately or clearly explained. Therefore, we have rewritten and expanded this section; the method of determining the bearing of the *compass run* is described now in the subsection “Identifying azimuths”.

We would like the authors to consider providing some form of analyses of how the result change with different speed and degree of turn.

We have expanded the description of the *turning point* and compass run, to provide a better understanding of the context in which changes in speed and direction occur. Speed within the *turning trajectory* is/was the slowest relative to other comparable segments of the excursion, consistent with this reflecting change in the underlying behaviour. However, this parameter was relative to the speed of each outbound track, and therefore, varied within and between dogs. Also, the degree of turn was relative; because the *compass run* generally coincided with the north-south axis, irrespective of the actual homeward azimuth (Figure 3), the turn angle depended on the deviation of the last outbound segment from the north-south axis. Therefore, we believe, a formal analysis of speed and turn angle may be uninformative and/or misleading given our methodology.

2) The argument that the "compass run" could be used to bring a cognitive map back into register was found to be vague and unclear, and the reviewers believed that there was no evidence for this. This interpretation should be revisited or additional data provided in favour for it.

We agree with the reviewers, though our decision to include this in the original submission was to stimulate an interesting discussion, or alternative perspectives, regarding the functional significance of north-south alignment in vertebrates, albeit very speculative. Therefore, we deleted two questioned paragraphs in the Discussion. In fact, the criticism corresponds also with the next point (3) raised by the reviewers.

3) A discordance between the Introduction and the Discussion needs to be addressed. The Introduction seems to focus on true navigation as the explanation for scouting, but the Discussion focuses on path integration and cognitive maps as the explanation, neither of which are true navigation by the standard definitions.

We deleted passages dealing with cognitive maps and path integration and shortened the Discussion to keep it concise and straightforward. The Introduction mainly focuses on what is known about navigation in dogs. Only one sentence mentions the possibility of dogs using scent trails or true navigation to home of large distances.